# Predictive Value of Flow Cytometry Quantification of BAL Lymphocytes and Neutrophils in ILD

**DOI:** 10.3390/cells13242066

**Published:** 2024-12-13

**Authors:** Erika M. Novoa-Bolivar, José A. Ros, Sonia Pérez-Fernández, José A. Campillo, Ruth López-Hernández, Rosana González-López, Almudena Otálora-Alcaraz, Cristina Ortuño-Hernández, Lourdes Gimeno, Inmaculada Ruiz-Lorente, Diana Ceballos-Francisco, Manuel Muro, Elena Solana, Pablo Martinez-Camblor, Alfredo Minguela

**Affiliations:** 1Immunology Service, Clinical University Hospital Virgen de la Arrixaca (HCUVA), Biomedical Research Institute of Murcia Pascual Parrilla (IMIB), 30120 Murcia, Spain; e.m.novoab@hotmail.com (E.M.N.-B.); josea.campillo@carm.es (J.A.C.); ruth.lopez2@carm.es (R.L.-H.); rosana13a@hotmail.com (R.G.-L.); almudena.o.a@gmail.com (A.O.-A.); cristina.ortunoh@um.es (C.O.-H.); lourdes.gimeno@carm.es (L.G.); irl_98@hotmail.com (I.R.-L.); dceballosf@gmail.com (D.C.-F.); manuel.muro@carm.es (M.M.); 2Pneumology Service, Clinical University Hospital Virgen de la Arrixaca (HCUVA), Biomedical Research Institute of Murcia Pascual Parrilla (IMIB), 30120 Murcia, Spain; jarl77@yahoo.es (J.A.R.); elenasolana90@gmail.com (E.S.); 3Department of Statistics and Operations Research and Mathematics Didactics, University of Oviedo, 33007 Asturias, Spain; perezsonia@uniovi.es; 4Department of Biomedical Data Science, Geisel School of Medicine at Dartmouth, 7 Lebanon Street, Suite 309, Hinman Box 7261, Hanover, NH 03755, USA; pablo.martinez-camblor@hitchcock.org

**Keywords:** flow cytometry, bronchoalveolar lavage, lymphocyte, neutrophil, lung diseases, stratification

## Abstract

Interstitial lung diseases (ILDs) are pathologies affecting the pulmonary interstitium and, less frequently, the alveolar and vascular epithelia. Bronchoalveolar lavage (BAL) is commonly used in ILD evaluation since it allows the sampling of the lower respiratory tract. The prognostic value of BAL cell counts in ILD is unknown. Flow cytometry quantification of lymphocytes and neutrophils in BAL of 1074 real-life consecutive patients were retrospectively correlated with clinical, radiological, anatomopathological, functional/spirometry, and evolutionary data. Cut-offs with predictive value were established at 7% and 5% for lymphocytes and neutrophils, respectively. Three risk stratification groups (Risk-LN) were established: FAVORABLE (lymphocytes > 7% and neutrophils < 5%), INTERMEDIATE (rest of patients), and UNFAVORABLE (lymphocytes < 7% and neutrophils > 5%), showing 75th percentile overall survival (OS) of 10.0 ± 1.4, 5.8 ± 0.6, and 3.0 ± 0.3 years (*p* < 0.001), respectively. A scoring model combining Risk-LN and the age of the patients with great predictive capacity for OS on fibrotic and non-fibrotic ILDs is proposed. This score is an independent predictive factor (HR = 1.859, *p* = 0.002) complementary to the fibrosis status (HR = 2.081, *p* < 0.001) and the type of treatment. Flow cytometry of BAL provides rapid and accurate quantification of lymphocytes and neutrophils, allowing the establishment of a risk score model that is useful in the clinical management of fibrotic and non-fibrotic ILDs from the time of diagnosis.

## 1. Introduction

Interstitial lung disease (ILD) is a group of heterogeneous pathologies with different etiologies and comorbidities that commonly involve the pulmonary interstitium and, less frequently, also the alveolar and vascular epithelium. Repair processes are involved in the disease, with varying degrees of inflammation and fibrosis [1,2]. Idiopathic pulmonary fibrosis (IPF) is the most lethal ILD. Clinical guidelines have defined several factors associated with worse prognosis in IPF, such as age, presence of symptoms, impaired lung function, greater severity of pulmonary fibrosis, or the presence of usual interstitial pneumonia (UIP). Scores have been established to stratify risk mortality in IPF; the most widely used is the GAP [3], which includes gender (G), age (A), and physiology (P) functional tests such as diffusing capacity of the lungs for carbon monoxide (DLCO) and forced vital capacity (FVC). The GAP score does not predict a drop in pulmonary function [4], nor does it take into account the presence of comorbidities, which has led to introducing other variables to adapt the score, such as the Charlson comorbidity index [5], changes in body mass index [6], the drop in oxygen saturation in the 6 min walk test [7,8,9], hospital admissions [5], or echocardiography data [10]. The GAP score has been adapted for the rest of the ILDs, such as the ILD-GAP [11], which has proven to be useful in different ILDs [12,13]. In progressive pulmonary fibrosis (PPF), age is a factor in poorer pulmonary function and mortality; furthermore, the degree of fibrosis is related to worse prognosis [14]. In patients with systemic sclerosis-associated interstitial lung disease (SSc-ILD), high-resolution computed tomography (HRTC) [15] and ultrasound [16] of the lung can help in the risk stratification of disease severity.

Some studies on ILD have evaluated the possible prognostic value of leukocyte quantification in bronchoalveolar lavage (BAL) [17,18], especially during acute processes [19,20]. Results have shown that higher percentages of neutrophils are associated with an increased risk of exacerbations [13], and of mortality during exacerbations [19,20], because neutrophils might be related to more alveolar damage and fibrosis. In contrast, higher percentages of lymphocytes are associated with better prognoses [19,20]. However, these results have not been translated into prognostic scores; therefore, the usefulness of BAL cytology remains limited to the diagnostic algorithm of some ILDs [21].

This retrospective study evaluates the prognostic value of BAL leukocyte subset quantification at diagnosis in a large series of real-life ILDs. The results show that neutrophil and lymphocyte counts have opposite prognostic values and can be used for risk stratification. The main novelty of our work is that leukocyte quantification is performed by flow cytometry, providing faster and more accurate results than conventional cytology, and it can be used, together with age and radiological patterns, to estimate a new, highly predictive survival score.

## 2. Materials and Methods

### 2.1. Patients and Samples

This retrospective and observational study from real-life medicine included clinical, radiological, and anatomopathological data from patients referred from public hospitals of Murcia Region, Spain. A total of 1483 consecutive BAL samples were analyzed by flow cytometry between 2000 and 2018. Exclusion criteria were lung or other cancers (n = 211), asthma (n = 86), chronic obstructive pulmonary disease (n = 82), or tuberculosis (n = 30). Finally, the study included 731 BAL samples from patients with ILD and 231 from patients with pulmonary infections at the moment of BAL. As control-BAL group, 112 BAL samples were included from patients with suspected lung disease (mainly due to the presence of micronodules on radiological images) that after years of follow-up did not show clinicopathological evidence of lung disease in the electronic medical record. Furthermore, to compare the outcomes of ILD patients, 243 patients with monoclonal gammopathy of undetermined significance (MGUS), without pulmonary disease and free of adverse prognostic factors [22], were included as a survival control of the general population.

The BAL sampling procedure was performed following the official American Thoracic Society clinical practice guidelines [21].

The end of evolutionary data collection was on 1 April 2023. Anamnesis, clinical examination, radiology (radiography and HRTC), BAL cytomorphology and microbiology, and anatomopathological and functional pulmonary studies were performed according to clinical practice in each hospital. Functional and spirometry data were available only in 182 out of 731 ILDs (24.89%). Pulmonary fibrosis was computed with the presence of reticular changes, traction bronchiectasis, and honeycombing in the radiological study. Diagnostic criteria of ILD subtype were based on the ATS/ERS classification [23]. Immunomodulatory treatment was administered according to standard practice [24]. Antifibrotics were available from 2014 onward, with limited access according to local restrictions. Based on information from the electronic medical record, patients were grouped into 3 treatment groups: (1) patients who did not require systemic immunomodulatory treatment; (2) patients who received only corticosteroids; and (3) patients who required other immunomodulatory treatments (rituximab, azathioprine, mycophenolate mofetil, cyclophosphamide, or tacrolimus) after corticosteroids, as described previously [25].

The institutional review board (IRB-00005712) approved the study. Written informed consent was obtained from all patients in accordance with the Declaration of Helsinki. 

### 2.2. Immunophenotype Studies

BAL samples from different hospitals in Murcia Region were transported refrigerated at 4–8 °C in isothermal containers and processed immediately in our laboratory upon receipt. Briefly, the volume and appearance of samples were recorded. All the volume was immediately centrifuged at 1800 rpm, the supernatant was aspirated, and the pellet was washed once with 15 mL of FACSFlow (Becton Dickinson; BD; San Jose, CA, USA). Finally, samples were resuspended in 0.5 mL of FACSFlow (BD) and 50 µL stained immediately in TruCount tubes (BD) with the following antibody mix: CD1a PE (HI149), CD3 BV510 (SK7), CD4 APC (SK3), CD8 PE-Cy7 (SK1), CD16 V450 (3G8), CD19 APC (SJ25C1), CD20 FITC (L27), CD45 APC-H7 (2D1), and HLA-DR PerCp (L243) from BD and CD66 FITC (Kat4c) from Dako (Santa Clara, CA, USA). Samples were vortexed and incubated for 10 min at room temperature in the dark, then lysed with ammonium chloride solution (BD) for 7 min, and a minimum of 0.5 million cells were acquired immediately in 8-color FACSCanto-II flow cytometer (BD). Photomultiplier (PMT) voltages were adjusted daily using CS&T beads (BD). Fluorescence compensations were adjusted using FC beads (BD) every two months and finely adjusted on a daily basis using negative events as reference for each fluorochrome [25,26].

DiVA^TM^ Software 9.0 (BD, San Jose, CA, USA) was used for analysis following the gating strategy described in Figure 1. No dead cell exclusion dyes were used; therefore, both alive and dead cells (identified by the loss in FSC and SSC) were included, as long as they maintained the expression of leukocyte markers, especially CD45 [25].

### 2.3. Statistical Analysis

As described previously [25], data were collected in Excel (Microsoft Corporation, Redmond, WA, USA) and analyzed in SPSS 21.0 (Armonk, NY, USA). Kaplan–Meier and Log-Rank tests were used for survival estimation. Overall survival (OS) was defined as the time from first BAL analysis to death, with living patients censused on the date of last follow-up. Outcome of patient groups expressed in years was estimated as the 75th-percentile-OS (75p-OS). Comparisons between qualitative variables were performed by the two-tailed Fisher’s exact test. Quantitative variables were compared using ANOVA and DMS post hoc tests. Receiver operating characteristic (ROC) analysis was used to explore patient OS and to determine the optimal cut-off for quantitative variables. Multivariate analysis of predictive factors for OS was performed using the Cox proportional hazards model (stepwise regression). Hazard ratio (HR) and 95% confidence interval were estimated. *p* < 0.05 was considered statistically significant.

## 3. Results

### 3.1. Clinical, Biological, and Therapeutic Characteristics of Study Groups

Table 1 shows the biological and clinical characteristics of the patient and control groups as well as the main treatments for each pulmonary pathology. 

Age and sex showed comparable values among the study groups (controls, ILDs, and infectious pulmonary diseases). Some ILD pathologies were predominantly observed among men, such as pneumoconiosis (96.3%), Eosinophilic-ILD (80.0%), IPF (71.7%), LIP (69.2%), and PLCH (66.7%). Comparable age was observed among the groups, although patients with PLCH were younger (36.7 years) compared with the age of the rest of the ILD patients (58.7 years). Connective tissue disease ILDs were not included as a group, since very few of these patients had a BAL study. It should also be noted that one-third of the patients did not require systemic immunosuppressive treatment.

### 3.2. Risk Stratification of ILDs Based on Lymphocyte and Neutrophil Counts in BAL (Risk-LN)

Table 2 shows the percentage and absolute counts of the main leukocyte subsets in the bronchoalveolar lavage of the main disease groups. In the BAL control group, macrophages represented 89.8% of the cellularity with a mean of 98.2 ± 4.6 macrophages/µL. ILDs with or without fibrosis showed higher mean values of lymphocytes (22.4 ± 1% and 20.8 ± 1.6%, respectively) and neutrophils (7.7 ± 0.5 and 8.5 ± 0.8%, respectively) than BAL controls. IPF and infectious diseases had higher values of neutrophils (16.9 ± 1.5% and 46.0 ± 1.8%, respectively, *p* < 0.001) than BAL controls and other ILDs with or without fibrosis.

ROC analysis revealed cut-offs of 7.0% and 5.0% for lymphocytes and neutrophils, respectively, with predictive value on the OS (see Figure 2a for details). Longer 75p-OS was observed in patients with lymphocytes > 7.0% (7.8 ± 0.8 vs. 3.9 ± 0.6 years, *p* < 0.001) and neutrophils < 5.0% (8.8 ± 0.8 vs. 4.2 ± 0.5 years, *p* < 0.001) (Figure 2b). The combination of lymphocyte and neutrophil cut-offs showed four groups with different survival rates (Figure 2c,d): patients with lymphocytes > 7% and neutrophils < 5% showed the longest 75p-OS (10.0 ± 1.4 years, *p* < 0.001), with values comparable to those of the general population (75p-OS not reached); patients with lymphocytes < 7% and neutrophils > 5% showed the shortest 75p-OS (3.0 ± 0.3 years); the rest of the patients showed intermediate values for 75p-OS (5.8 ± 0.6 years). Therefore, an accurate counting of lymphocytes and neutrophils by flow cytometry in the BAL of patients with ILD and infectious diseases can provide a survival risk stratification.

Risk-LN predictive capacity was maintained in both ILD (10.0 ± 1.4, 5.7 ± 0.6, and 2.2 ± 0.4 years, *p* < 0.001) and infectious pulmonary diseases (not reached, 6.1 ± 0.6, and 4.0 ± 0.4 years, *p* < 0.01) with comparable 75p-OS values for favorable, intermediate, and unfavorable risk groups, respectively (Figure 3).

Because the analyzed samples came not only from our hospital but also from other hospitals, we tested whether the Risk-LN stratification could be influenced by transport conditions or processing time. Results showed very similar 75p-OS for local samples, which were processed in less than 30 min, and samples from other hospitals, which were processed in 60 to 90 min, from patients with favorable (12.0 ± 1.93 vs. 9.1 ± 1.69 years), intermediate (5.9 ± 1.43 vs. 5.2 ± 1.03 years), or unfavorable (3.3 ± 0.86 vs. 2.5 ± 0.88 years) Risk-LN stratifications (Appendix A).

### 3.3. Uneven Distribution of Risk-LN Groups Among Pulmonary Pathologies

The Risk-LN groups had an uneven distribution among the pulmonary pathologies (Figure 4). Favorable and intermediate groups were mostly represented in ILDs with a predominance of lymphocytes (sarcoidosis, HP, COP, and LIP), whereas the unfavorable group was mostly represented in pathologies with a predominance of neutrophils (infectious pulmonary disease, AIP, BR-ILD, and UIP). In pathologies with a predominance of macrophages, eosinophils, or Langerhans cells (DIP, NSIP, pneumoconiosis, PLCH, Eosinophilic-ILD, and U-ILD), the distribution of the Risk-LN groups was more even.

### 3.4. Risk-LN Stratification Preserves Its Predictive Value in Pathologies With or Without Pulmonary Fibrosis

Figure 5a shows the distribution of the most frequent radiological patterns in the different lung pathologies. As expected, the radiological patterns associated with pulmonary fibrosis (presence of reticular changes, traction bronchiectasis, and honeycombing) presented lower 75p-OS (2.9 ± 1.3 years) than nodular (not reached), alveoli interstitial (8.3 ± 0.6 years), or the rest of the patterns (11.4 ± 1.1 years) (Figure 5b). However, Risk-LN stratification maintained its prognostic value in patients without fibrosis (16.0 ± 1.5, 9.0 ± 0.7, and 6.4 ± 0.9 years, *p* = 0.014) or with fibrosis (5.1 ± 1.3 vs. 2.9 ± 0.6, and 1.9 ± 1.3 years, *p* < 0.005), respectively, for the favorable, intermediate, and unfavorable risk groups.

### 3.5. Risk-LN Predictive Capacity Was Independent of the Type of Treatment

Figure 6a shows patient outcomes according to the type of systemic therapy. Patients that did not receive systemic therapy showed longer 75p-OS (10.1 ± 1.4, *p* < 0.001) than those treated with other immunosuppressants (6.3 ± 1.3 years) or corticosteroids (4.3 ± 0.86 years).

Nonetheless, Risk-LN stratification maintained its predictive value in patients without systemic treatment (11.5 ± 0.98, 7.7 ± 0.76, and 5.8 ± 1.3 years, *p* < 0.05) or those treated with systemic corticosteroids (10.0 ± 2.5, 4.4 ± 2.1, and 1.7 ± 0.4 years, *p* < 0.001) or other immunosuppressants (9.9 ± 2.5, 6.4 ± 1.3, and 2.5 ± 0.7 years, *p* < 0.03) for the favorable, intermediate, and unfavorable risk groups.

### 3.6. Risk-LN, DLCO, and Fibrosis Are Independent and Complementary Predictive Factors in ILD

Appendix A shows the functional and spirometry parameters in the different ILDs. Although DLCO data were available in only 25.1% of patients, the ROC analysis revealed a cut-off of 55.5% for DLCO with predictive value on the OS. Patients with DLCO < 55.5% showed shorter 75p-OS (3.0 ± 0.6 vs. 9.1 ± 1.2 years, *p* < 0.001) than patients with DLCO > 55.5%. Nonetheless, Risk-LN could still identify patients with unfavorable risk among patients with DLCO > 55.5%, who showed shorter 75p-OS (4.1 ± 0.9 years vs. not reached, *p* < 0.001) compared with patients with favorable or intermediate risk. In patients with DLCO < 55.5%, unfavorable Risk-LN was associated with shorter 75p-OS (2.0 ± 0.9 vs. 6.3 ± 0.7 and 3.9 ± 0.8 years, *p* < 0.05) compared with patients with favorable or intermediate risk, respectively (Appendix A).

Risk-LN stratification was an independent predictive factor for OS (HR = 1.619, *p* = 0.008) that complemented the other independent predictive factors in ILDs, such as age (HR = 1.046, *p* = 0.001), fibrosis (HR = 2.048, *p* = 0.015), and DLCO (HR = 0.979, *p* = 0.001) (Appendix A).

### 3.7. Risk-Scoring Model for Overall Survival Useful in Fibrotic and Non-Fibrotic ILDs

Finally, a risk score model (Figure 7a) with predictive capacity for OS in fibrotic and non-fibrotic ILDs was proposed combining the Risk-LN (0, favorable; 1, intermediate; and 2, unfavorable) and the age range of patients (0, <50 years; 1, 50–70 years; and 2, >70 years). Decreasing 75p-OSs were observed in patients with 0, 1, 2, 3, and 4 scores, in both non-fibrotic (not reached, 16.0 ± 1.2, 7.3 ± 1.0, 3.1 ± 1.3, and 1.1 ± 0.2 years, *p* < 0.001) and fibrotic (10.0 ± 3.0, 6.9 ± 0.6, 2.8 ± 0.6, 1.3 ± 0.4, and 1.1 ± 0.2 years, *p* < 0.001) pulmonary pathologies. This OS risk score was an independent predictive factor (HR = 1.859, *p* = 0.002) complementary to the fibrosis status (HR = 2.081, *p* < 0.001) and the type of treatment (Figure 7b).

A quick and automated estimate of the risk score can be obtained at “https://bal-ildcalculator.imib.es/”. In addition, a diagnostic approach to the most probable pathology associated with that cellular pattern will be offered by applying logistic regression models to BAL leukocyte subsets of the patient. Examples of real cases are shown in Appendix A.

## 4. Discussion

Several models estimating the mortality risk in IPF [3] or general ILD [27] have been proposed to guide the clinical management of these patients. Models have taken into consideration clinical symptoms, pulmonary function, the extent of UIP, and even the Charlson comorbidity index score (CCIS) (ILD-GAPC) [27]. However, the immune cells infiltrating pulmonary tissue that are involved in lung deterioration and/or the fibrotic process have not been tested in these models. Our results show that lymphocytes and neutrophils, the inflammatory leukocytes that most frequently infiltrate the lung in ILD, play opposing roles in the clinical evolution of patients. While the former seem to protect, neutrophils are associated with a more accelerated progression of the disease and shorter patient survival. Flow cytometry of BAL can offer a quick and accurate quantification of these leukocyte subsets, revealing cut-offs with predictive capacity in ILD and thus allowing the establishment of a risk stratification based on the number of lymphocytes and neutrophils (Risk-LN). Risk-LN stratification can help to identify patients with a very unfavorable risk from the time of diagnosis as well as patients who have favorable risk and survival rates similar to the general population (without lung pathology), and therefore, who would not need close surveillance or therapies in the short term.

Furthermore, the Risk-LN stratification combined with the age range of patients allowed for the development of a “scoring” model that is able to predict the survival of patients with great precision in both non-fibrotic and fibrotic ILDs. With this predictive model, patients with very favorable long-term survival estimates—longer than 16 and 10 years in patients without and with pulmonary fibrosis, respectively—can be identified from diagnosis with the first BAL sample. In contrast, patients with very poor long-term overall survival estimates—shorter than 1.1 years—were also identified in both non-fibrotic and fibrotic ILDs. Unfortunately, the unavailability of functional and spirometry tests in some hospitals in our region between 2000 and 2015 has prevented the inclusion of the independent prognostic factor DLCO in the estimation of the risk score. Nonetheless, the score system described in this manuscript is simple and could provide a useful tool for guiding treatment decisions in the most common lung pathologies (quick score estimation is accessible to pulmonologists worldwide at https://bal-ildcalculator.imib.es/. Even so, independent studies in more recent patients will be necessary to determine the contribution of DLCO to the score model proposed in this study.

Although the protective effect of lymphocytes [20] and the harmful effect of neutrophils infiltrating the lung [17,18,20,28,29,30,31], or even in the peripheral blood [32,33,34], has been well stablished, the inclusion of lymphocyte and neutrophil quantification in clinical practice is still far from happening. Different reasons may be delaying the inclusion of these parameters in the management of lung pathology: first, the use of prognostic markers outside FPI is still unusual among pulmonologists; second, in real-life medicine, BAL cytology reports do not always include the number of leukocyte subsets, and if they do, the results may be delayed for several weeks. In this sense, flow cytometry can offer accurate results in less than an hour. However, flow cytometry continues to be a methodology with low penetration in pulmonology, although it has become an essential tool in many other clinical disciplines [35]. Technology improvement has facilitated the use of antibody panels large enough to correctly discriminate the different leukocyte subpopulations, thus clearly differentiating neutrophils from eosinophils, which can be difficult with panels with fewer antibodies. Nowadays, accessibility to this type of analysis has become widespread in most large hospitals, with professionals perfectly qualified to study BAL. Furthermore, the analysis strategy shown in this study, including the dead cells, can facilitate the accessibility of this technology in smaller hospitals that do not have onsite flow cytometry, since the Risk-LN predictive value was maintained even in samples that needed to be transported and were processed later than 60 to 90 min after extraction. Furthermore, our results support recent findings indicating that flow cytometry may be a good alternative to cytopathological analysis, as it provides comparable data when analyzing major leukocyte subsets, including lymphocytes and neutrophils [36].

The harmful effect shown by neutrophils in our series and in others [37], in the absence of the protective effect of lymphocytes, could serve as a guide for clinicians to establish early preventive therapies to mitigate pulmonary deterioration, particularly in patients with unfavorable Risk-LNs (predominance of neutrophils over lymphocytes) and short survival estimates (scores higher than 2 in non-fibrotic patients and scores higher than 1 in fibrotic patients). Although Risk-LN stratification preserved its predictive value in patients without treatment and in patients with different types of immunosuppressive therapies (corticosteroids or other immunosuppressants), it would have been interesting to evaluate its predictive capacity in patients with the new antifibrotic treatments [38,39,40]. However, just a minority of patients from our series received this therapy; therefore, this should be addressed in future studies.

In conclusion, although the predictive capacity of the Risk-LN score should be confirmed in larger prospective studies to evaluate its prognostic value, and it should incorporate other variables with prognostic value such as DLCO, our results clearly show that flow cytometry of BAL can offer predictive information independent of DLCO and the fibrotic status with potential utility for treatment personalization in ILD from the time of diagnosis.

## Figures and Tables

**Figure 1 cells-13-02066-f001:**
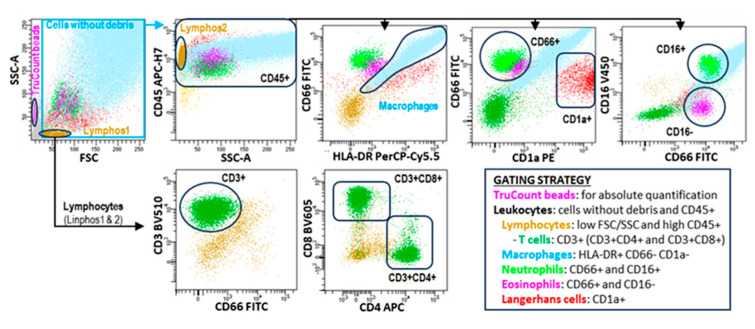
Flow cytometry analysis of the leukocyte subsets contained in BAL samples. Leukocyte subsets were identified following the hierarchical and logical gating strategy shown in the figure. TruCount beads were used to calculate the absolute cell counts per microliter following the manufacturer’s instructions.

**Figure 2 cells-13-02066-f002:**
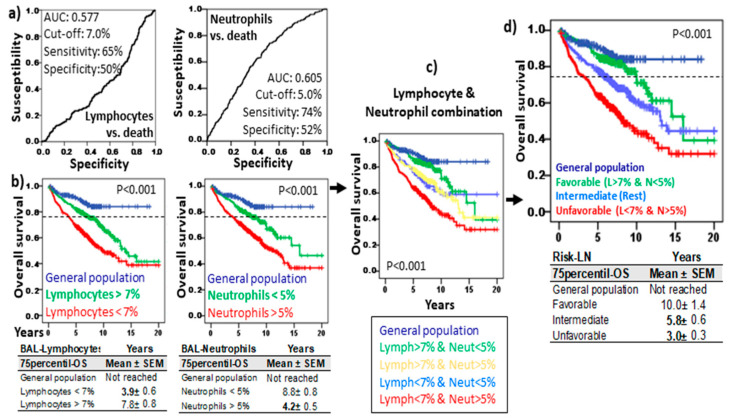
Risk stratification of ILD and infectious disease based on lymphocyte and neutrophil counts of BAL. (**a**) Receiver operating characteristic curve (ROC) of lymphocytes and neutrophils related to overall survival (OS). Area under the curve (AUC), cut-off, and the highest sensitivity and specificity are shown. (**b**) Kaplan–Meier and Log-Rank tests for OS of ILD and infectious disease patients with lymphocytes and neutrophils below and over their respective cut-offs and OS of general population (without lung diseases). (**c**) Kaplan–Meier and Log-Rank tests for OS of ILD patients according to the combinations of lymphocytes (above or below 7%) and neutrophils (above or below 5%). (**d**) Kaplan–Meier and Log-Rank tests for OS of ILD patients according to the Risk-Lymphocyte/Neutrophil stratification (Risk-LN): favorable (lymphocyte > 7% and neutrophil < 5%), unfavorable (lymphocyte < 7% and neutrophil > 5%), and intermediate (rest). The 75th percentile OS (Mean ± SD) are shown for the patients and general population in each section.

**Figure 3 cells-13-02066-f003:**
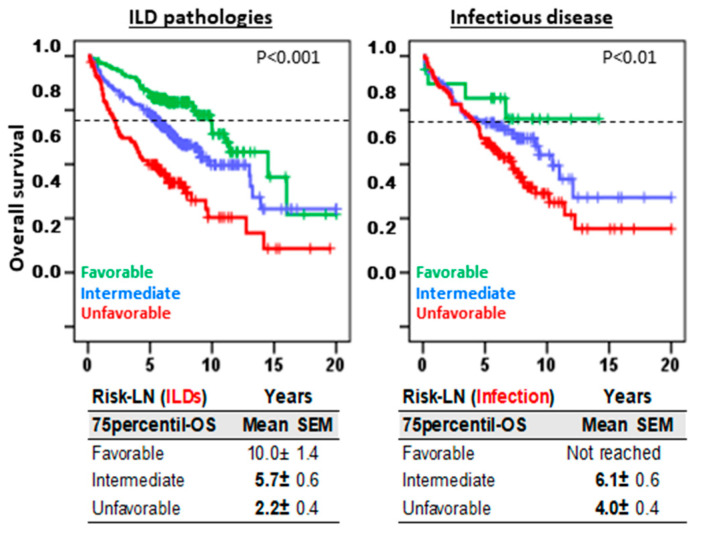
Risk-Lymphocyte/Neutrophil (Risk-LN) stratification offers predictive information in both ILDs and infectious diseases. Kaplan–Meier and Log-Rank tests for overall survival (OS) according to Risk-LN in patients with ILD pathologies and in patients with infectious disease. The 75th-percentile-OS is shown.

**Figure 4 cells-13-02066-f004:**
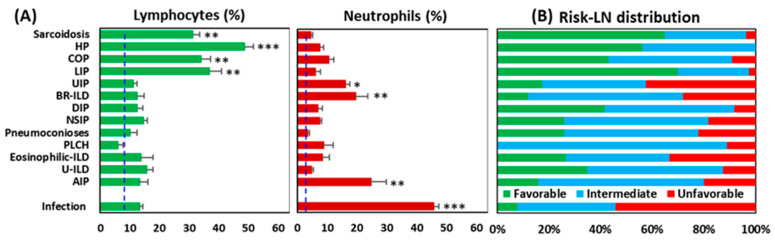
Distribution of Risk-Lymphocyte/Neutrophil (Risk-LN) stratification groups among pulmonary pathologies. (**A**) Percentage of lymphocytes and neutrophils in the flow cytometry analysis of BAL samples from patients with different pulmonary pathologies. * *p* < 0.05, ** *p* < 0.01, *** *p* < 0.001 in the ANOVA and DMS post hoc tests. The dashed lines indicate the mean value of lymphocytes (6.5%) and neutrophils (2.12%) in the control BAL group. (**B**) Distribution of Risk-LN groups among pulmonary pathologies: favorable (lymphocyte > 7% and neutrophil < 5%), unfavorable (lymphocyte < 7% and neutrophil > 5%), and intermediate (rest).

**Figure 5 cells-13-02066-f005:**
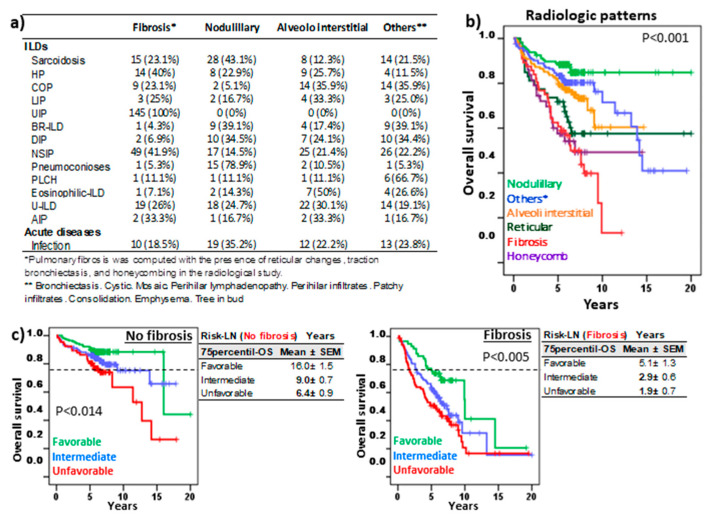
Risk-Lymphocyte/Neutrophil stratification (Risk-LN) in ILD patients with pulmonary fibrosis. (**a**) Frequency of the most common lung imaging patterns among ILD subtypes. (**b**) Kaplan–Meier and Log-Rank tests for overall survival (OS) of ILD patients according to their predominant lung pattern. * other radiological patterns. (**c**) Kaplan–Meier and Log-Rank tests for OS of ILD patients with or without fibrosis according to Risk-LN stratification. The 75th-percentile-OS is shown in each case.

**Figure 6 cells-13-02066-f006:**
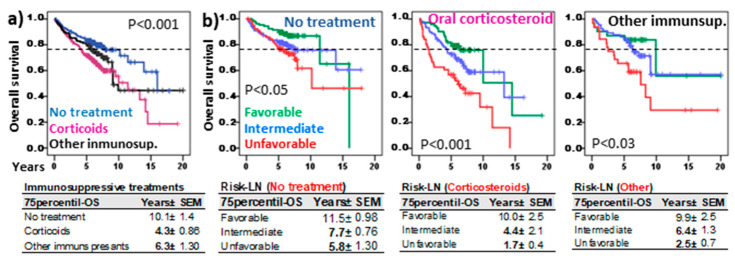
Risk-Lymphocyte/Neutrophil (Risk-LN) stratification in ILD patients with different systemic immunosuppressive treatments. (**a**) Kaplan–Meier and Log-Rank tests for overall survival (OS) of ILD patients according to the type of treatment: no systemic treatment, corticosteroids, or other immunosuppressants (rituximab, azathioprine, mycophenolate mofetil, cyclophosphamide, tacrolimus). (**b**) Kaplan–Meier and Log-Rank tests for OS of ILD patients according to the Risk-Lymphocyte/Neutrophil stratification (Risk-LN) and the type of treatment. The 75th-percentile-OS is shown in each case.

**Figure 7 cells-13-02066-f007:**
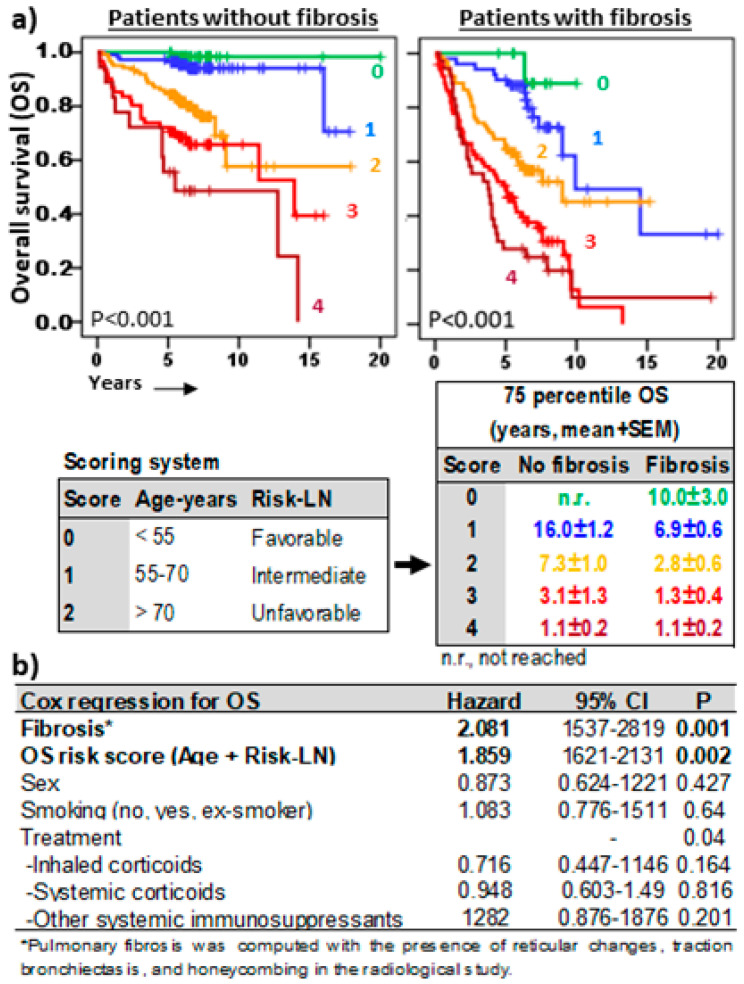
Overall survival risk scoring systems in ILDs. (**a**) Kaplan–Meier and Log-Rank tests for overall survival (OS) of ILD patients according to a scoring model including age range (<55 years = 0, 55–70 years = 1, >55 years = 2) and Risk-Lymphocyte/Neutrophil stratification (Risk-LN) (favorable = 0, intermediate = 1, unfavorable = 2). The 75th-percentile-OS was estimated separately for patients with or without pulmonary fibrosis. (**b**) Cox regression analysis for OS in ILD patients for sex, type of ILD, smoking status, type of treatment, fibrosis status, and the OS risk score.

**Table 1 cells-13-02066-t001:** Biological, clinical, and therapeutic characteristics of patient and control groups.

	Patients	Sex	Age	Fibrosis ^1^	Treatments (%) ^2^		
	(N)	(% man)	(Mean ± SD)	(%)	NT	SC	Other
**Control groups**	**355**	**53.2%**	**61.2 ± 13.5**	-			
General population ^3^	243	53.1%	67.9 ± 12.3	-			
BAL control ^4^	112	53.3%	53.5 ± 16.6	0%			
**Interstitial lung diseases (ILD)**	**731**	**58.1%**	**58.7 ± 16.4**	**38.2%**	**32**	**32**	**18**
Sarcoidosis	82	48.20%	55.6 ± 14.8	19.5%	34	38	15
Hypersensitivity pneumonitis (HP)	48	49.0%	50.4 ± 17.6	29.1%	13	46	13
Organized cryptogenic pneumonia (COP)	44	46.50%	62.3 ± 17.4	18.1%	18	52	21
Lymphocytic interstitial pneumonia (LIP)	37	69.20%	53.2 ± 15.9	8.1%	49	24	16
Usual interstitial pneumonia (UIP) ^5^	145	71.70%	65.8 ± 12.3	100%	25	35	19
Respiratory bronchiolitis ILD (RB-ILD)	25	40.0%	53.3 ± 22.8	8.0%	36	36	8
Desquamative interstitial pneumonitis (DIP)	35	52.8%	54.4 ± 20.8	5.7%	46	23	17
Nonspecific interstitial pneumonia (NSIP)	156	48.1%	59.3 ± 15.5	33.9%	31	28	30
Pneumoconiosis	27	96.3%	56.5 ± 16.0	18.5%	42	23	4
Pulmonary Langerhans cell histiocytosis (PLCH)	9	66.7%	36.7 ± 15.8	11.1%	67	11	11
Eosinophilic-ILD	17	80.0%	48.4 ± 22.7	5.9%	0	76	6
Unclassifiable-ILD (U-ILD)	80	60.0%	61.3 ± 12.4	25.0%	49	15	10
Acute interstitial pneumonia (AIP)	26	56.0%	57.5 ± 21.9	15.3%	29	24	12
**Pulmonary infectious diseases**	**231**	**59.0%**	**62.1 ± 15.6**	**15.1%**	**27**	**22**	**18**

^1^ Pulmonary fibrosis was computed with the presence of reticular changes, traction bronchiectasis, and honeycombing in the radiological study. ^2^ Immunomodulatory treatment was administered according to standard practice [24]: NT: no treatment; SC: systemic corticosteroid; Other: rituximab, azathioprine, mycophenolate mofetil, cyclophosphamide, or tacrolimus. ^3^ Patients with monoclonal gammopathy of undetermined significance (MGUS), without pulmonary disease and free of adverse prognostic factors [22]. ^4^ BAL performed for etiological affiliation, but during the follow-up, no ILD pathology was evident. ^5^ Following ATS/ERS diagnostic criteria, idiopathic pulmonary fibrosis was included in the UIP group.

**Table 2 cells-13-02066-t002:** Percentage and absolute counts of the main leukocyte subsets in the bronchoalveolar lavage of the main disease groups.

	BAL Control		ILD Without Fibrosis		ILD with Fibrosis		IPF		Infectious Diseases	
	(n = 112)		(n = 452)		(n = 279)		(n = 145)		(n = 231)	
	%	Cels/µL	%	Cels/µL	%	Cels/µL	%	Cels/µL	%	Cels/µL
Lymphocytes	6.3 ± 0.6	6.9 ± 0.7	22.4 ± 1.0 *	24 ± 1.2 *	20.8 ± 1.6 *	22.7 ± 2.0 *	10.5 ± 1.2 *	11.1 ± 1.4 *	13 ± 1.1 *	14.6 ± 1.3 *
Macrophages	89.8 ± 0.7	98.2 ± 4.6	65.9 ± 1.1 *	70.1 ± 1.5 *	67.1 ± 1.8 *	72.9 ± 3.2 *	69.1 ± 1.8 *	71.8 ± 2.8 *	37.9 ± 1.5 *	44.5 ± 2.6 *
Neutrophils	2.1 ± 0.2	2.5 ± 0.4	7.7 ± 0.5 *	8.8 ± 1.0 *	8.5 ± 0.8 *	11.1 ± 2.4 *	16.9 ± 1.5 *	18.2 ± 1.8 *	46.0 ± 1.8 *	62.7 ± 7.1 *
Eosinophils	0.4 ± 0.1	0.5 ± 0.2	1.6 ± 0.3	1.6 ± 0.3	1.3 ± 0.2	1.6 ± 0.4	1.1 ± 0.2	1.1 ± 0.2	0.9 ± 0.1	1.1 ± 0.2
CD1a + D.C.	0.5 ± 0.1	0.6 ± 0.1	0.7 ± 0.1	0.7 ± 0.1	0.7 ± 0.1	0.8 ± 0.1	0.9 ± 0.1	1.1 ± 0.1	0.5 ± 0.1	0.7 ± 0.1
CD4 + Lymphs	2.8 ± 0.3	3.0 ± 0.4	10.8 ± 0.6	11.5 ± 0.7	10.4 ± 1.0	11.0 ± 1.1	5.6 ± 0.8	5.9 ± 0.9	6.3 ± 0.7	7.1 ± 0.8
CD8 + Lymphs	2.5 ± 0.3	2.7 ± 0.3	8.8 ± 0.5	9.5 ± 0.7	7.7 ± 0.9	8.6 ± 1.1	3.5 ± 0.4	3.8 ± 0.6	4.7 ± 0.5	5.3 ± 0.6

D.C., Langerhans dendritic cells; ILD, interstitial lung disease; IPF, idiopathic pulmonary fibrosis. * *p* < 0.001 in the ANOVA and DMS post-hoc tests compared to BAL-control groups.

## Data Availability

The data that support the findings of this study are available from the corresponding author upon reasonable request.

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
