# Peer review of "Predictive Value of Flow Cytometry Quantification of BAL Lymphocytes and Neutrophils in ILD"

_cells, 2024, doi:10.3390/cells13242066_

Round 1
Reviewer 1 Report
Comments and Suggestions for Authors
The manuscript is quite well written. The topic is intersting. I have some comments:
1) Abstract. Flow cytometry of BAL offers a quick and accurate quantification of lymphocytes and neutrophils, useful in the clinical management of different ILDs from the time of diagnosis. Abstract might be beneficial to include a sentence that briefly summarizes the key findings of the study. This can provide readers with a quick overview of the research.
2) 1. Introduction 32 Interstitial lung disease (ILD) is a group of heterogeneous pathologies with different 33 aetiologies and comorbidities that commonly involve the pulmonary interstitium and, 34 less frequently, also the alveolar and vascular epithelium. Repair processes are involved 35 in the disease, with varying degrees of inflammation and fibrosis [1, 2]. Idiopathic pul-36 monary fibrosis (IPF) is the most lethal ILD. Clinical guidelines have defined several 37 factors associated with worse prognosis in IPF, such as age, presence of symptoms, im-38 paired lung function, greater severity of pulmonary fibrosis or the presence of usual 39 interstitial pneumonia (UIP). Scores have been established to stratify risk mortality in 40 IPF, the most widely used is the GAP [3] including gender (G), age (A) and physiology 41 (P) functional tests, such as diffusing capacity of the lungs for carbon monoxide (DLCO) 42 and forced vital capacity (FVC). Authors are kindly requested to emphasize the current concepts about these issues in the context of recent knowledge and the available literature. These articles should be quoted in the References list. References
1. High-Resolution Computed Tomography: Lights and Shadows in Improving Care for SSc-ILD Patients. Diagnostics (Basel). 2021;11(11):1960. Published 2021 Oct 22. doi:10.3390/diagnostics11111960
2. The role of ultrasound in systemic sclerosis: On the cutting edge to foster clinical and research advancement. J Scleroderma Relat Disord. 2021;6(2):123-132. doi:10.1177/2397198320970394
3) This retrospective study evaluates the prognostic value of BAL leukocyte subset 60 quantification at diagnosis in a large series of real-life ILDs. Results show that neutrophil 61 and lymphocyte counts have opposite prognostic value and can be used for risk stratifi-62 cation. Unlike previous studies, quantification is performed by flow cytometry, provid-63 ing faster and more accurate results than conventional cytology, and it can be used, to-64 gether with age and radiological patterns, to estimate a new highly predictive survival 65 score. I suggest to underline the novelty of the study.
4) 2. Materials and Methods 67 2.1. Patients and samples 68 This retrospective and observational study from real-life medicine included clinical, 69 radiological, and anatomopathological data from patients referred from public hospitals 70 of Murcia Region, Spain. A total of 1483 consecutive BAL samples were analyzed by flow 71 cytometry between 2000 and 2018. Samples from patients with lung or other cancers 72 (n=211), asthma (n=86), chronic obstructive pulmonary disease (n=82) or tuberculosis 73 (n=30) were excluded. Finally, the study included 731 BAL samples from patients with 74 ILD and 231 from patients with pulmonary infections at the moment of BAL. As con-75 trol-BAL group, 112 BAL samples were included from patients with suspected lung dis-76 ease (mainly due to the presence of micronodules on radiological images) that after years 77 of follow-up did not show clinicopathological evidence of lung disease in the electronic 78 medical record. Besides, to compare the outcome of ILD patients, 243 patients with 79 monoclonal gammopathy of undetermined significance (MGUS), without pulmonary 80 disease and free of adverse prognostic factors [20], were included as a survival control of 81 general population. I suggest to underline the exclusion and the inclusion criteria.
5) 3. Results 148 3.1. Clinical, biological and therapeutic characteristics of study groups .... I suggest to underline the most important results to clarify the conclusions.
6) 4. Discussion 327 Several models estimating the mortality risk in IPF [3] or general ILD [24] have been 328 proposed to guide the clinical management of these patients. Models have taken into 329 consideration clinical symptoms, pulmonary function, the extent of UIP and even the 330 Charlson comorbidity index score (CCIS) (ILD-GAPC) [24]. However, the immune cells 331 infiltrating pulmonary tissue that are involved in lung deterioration and/or the fibrotic 332 process have not been tested in these models. The discussion section needs to be improved. It is necessary to clarify the results obtained and compare them with previous or similar studies.
Author Response
Response 1:The conclusion of the abstract has been changed to provide a more informative message. In our opinion the abstract already contains the main information described in the manuscript.
Response 2: We have included a new paragraph to summarize and to include these references “In patients with systemic sclerosis-associated interstitial lung disease (SSc-ILD), high-resolution computed tomography (HRTC) [15] and ultrasound [16] of the lung can help in the risk-stratification of disease severity”.
Response 3: Thank you very much, we have done so.
Response 4: Thank you very much, we have done so.
Response 5: We have included this paragraph: “Age and sex showed comparable values among study groups (controls, ILDs and in-fectious pulmonary diseases). Some ILD pathologies were predominantly observed among men, such as pneumoconiosis (96.3%), Eosinophilic-ILD (80.0%), IPF (71.7%), LIP (69.2%), and PLCH (66.7%). Comparable age was observed among groups, although pa-tients with PLCH were younger (36.7 years) compared to the age of the rest of ILD patients (58.7 years). Indicate that, connective tissue disease-ILDs were not included as a group, since very few of these patients had BAL study. It should also be noted that one third of the patients did not require systemic immunosuppressive treatment.”
Response 6: In the third paragraph of the discussion, lines 343-346, we describe that “Although the protective effect of lymphocytes [20] and the harmful effect of neutrophils infiltrating the lung [17,18,20,27–30], or even in peripheral blood [31–33], has been well stablished, the inclusion of lymphocyte and neutrophil quantification in the clinical practice is still far from happening.”. These references summarize the current knowledge of the role of neutrophils and lymphocytes in the development of fibrosis and the patient outcome in pulmonary pathology. We believe that it is not necessary to extend the discussion further, which as you will have seen is focused on highlighting the usefulness that flow cytometry can provide in the clinical management of these patients.
Reviewer 2 Report
Comments and Suggestions for Authors
Novoa-Bolivar and colleagues use flow cytometry to quantify lymphocytes and neutrophils in the BAL of patients with ILD and propose that these cells have prognostic value. The manuscript, as it stands, has significant issues, particularly in the methodological section, which impacts data interpretation.
1. Did the authors assess the cellularity of the BAL? How many cells were present on average? Was a cytospin performed to quantify the cells? Was the viability of the BAL cells assessed? This information is crucial to understand if the different samples are comparable to each other.
2. Regarding the BAL, what does the term "refrigerated" mean? Were the samples not analyzed immediately after collection? The authors should be more precise in describing how the samples were processed before flow cytometric analysis.
3. Why did the authors choose to analyze the BAL instead of blood, which is a more easily obtainable sample and requires less invasive procedures?
4. Did the patients with gammopathies undergo BAL as well? The methods do not clarify which samples were analyzed in these patients.
5. The first paragraph of the methods section (describing the sample sizes of the different groups) is difficult to understand. The authors could present this information using a flowchart to simplify comprehension. The same applies to the different markers in flow cytometry: a table with the markers and their biological significance would help the reader understand how the various cell populations were selected.
6. The results section lacks a clinical characterization of the patients. Moreover, the flow cytometry results are not mentioned. How did the authors determine the cutoffs for lymphocytes and neutrophils?
7. Figure 1, as presented, is incomprehensible. The labels in the plots are not legible. The authors could place this figure in the online data supplement and explain it step by step.
8. In general, the methods section is hard to read: the images are too small and unclear. The authors could simplify the results by choosing the most significant images.
Author Response
Response 1: All this information is clearly explained in the methods of the manuscript. Flow cytometry does not use cytospin, this is for microscopic cytologic analysis. The normal process is as simple as centrifuge the total volume remitted to our laboratory, wash once with FACSFlow, resuspend in 500µl and them stain 50µl of the concentrated samples. As described in the manuscript we use TruCount Tubes (Becton Dickinson) in all samples. So, knowing the initial volume, the dilutions factor, the volume stained (fix to 50µl) and the TruCount beads detected in the analysis, it is possible to obtain the absolute count of each cell subset just using “rule of three” following the manufacturer instructions. This is a process similar as to the one used for absolute quantification of peripheral blood lymphocyte subsets.
We clearly stated in the methods that “No dead cell exclusion dyes were used; therefore, both alive and dead cells (identified by the loss in FSC and SSC) were included, as long as they maintained the expression of leukocyte markers, especially CD45.” Lines 121-123.
We did estimate ROC curves and predictive score with both %s and Absolute counts and have good results with both methods. So, we decided to publish only the % values as they are much easier and cheaper to obtain for other flow cytometry labs. I think that the proportion of Neu/Lymph is the critical parameter in the lung environment.
Any way we have included a new table-2 describing the main leukocyte subpopulations for the main patients’ groups.
Response 2: The samples were processed immediately, but when they came from other hospitals they were transported in isothermal containers to 4-8ºC. We have made this clearer in the manuscript.
Response 3: Cytologic analysis of BAL is part of the diagnostic algorithm for many interstitial lung diseases to evaluate pulmonary infiltrating leukocytes. The distribution of BAL leukocyte subpopulations is very different in each type of lung disease and therefore its study has great diagnostic utility. In this work we demonstrate that, in addition, it also has great predictive value. Besides, peripheral blood cells do not reflect the inflammatory process taking place in the lung and therefore they are not informative.
Response 4: In lines 81-82 we clearly state that these patients “were included as a survival control of general population”. The only data used from these patients was their evolutionary follow-up over 10-15 years, in order to know their survival. But of course, they DID NOT undergo BAL since they had no pulmonary pathologies. As described in the paper, our purpose was to know the predictive capacity of the scoring system described in this work. It is not easy to have long-term survival data to perform Kaplan-Meier analysis in a healthy population. However, we had such information available in patients with benign gammopathies with good prognosis and without pulmonary pathology and with an age comparable to patients with pulmonary pathology. These patients with gammopathy were very useful to confirm that patients with good prognostic pulmonary pathology (low neutrophils and high lymphocytes) had survival rates comparable to the population without pulmonary pathology.
Response 5: We have slightly rephrased the paragraph describing the sample sizes of the different groups. Hope it is clearer. Besides we have made COMPLETELY NEW the figure describing the gating strategy followed in the flow cytometry analysis. Hope this is also clearer.
Response 6: We have included this paragraph: “Age and sex showed comparable values among study groups (controls, ILDs and in-fectious pulmonary diseases). Some ILD pathologies were predominantly observed among men, such as pneumoconiosis (96.3%), Eosinophilic-ILD (80.0%), IPF (71.7%), LIP (69.2%), and PLCH (66.7%). Comparable age was observed among groups, although pa-tients with PLCH were younger (36.7 years) compared to the age of the rest of ILD patients (58.7 years). Indicate that, connective tissue disease-ILDs were not included as a group, since very few of these patients had BAL study. It should also be noted that one third of the patients did not require systemic immunosuppressive treatment.”
You are correct that a detailed description of the cellularity found in BALs from these patients was lacking, and so Table-2 and a brief description of the same has been added. This will help to understand the cut-offs and why lymphocytes and neutrophils were selected as the main leukocyte subpopulations to establish a predictive score. The lymphocyte and neutrophil cut-offs, we believe, are sufficiently explained in Figure 2 in section 3.2, and do not need further explanation.
Response 7: As we have said, this figure has been made new from scratch.
Response 8: We think now it is clearer.
Round 2
Reviewer 1 Report
Comments and Suggestions for Authors
The authors adequately answered my questions. They edited the manuscript and took my suggestions into account. In my opinion this improved the manuscript. I have no further comments.
Reviewer 2 Report
Comments and Suggestions for Authors
Dear editor, the manuscript has been significantly improved, and the authors have addressed every comment. As far as I'm concerned, it can be accepted in its current form.